# Cleaning Strategies of Synthesized Bioactive Coatings by PEO on Ti-6Al-4V Alloys of Organic Contaminations

**DOI:** 10.3390/ma16134624

**Published:** 2023-06-27

**Authors:** Avital Schwartz, Alexey Kossenko, Michael Zinigrad, Viktor Danchuk, Alexander Sobolev

**Affiliations:** 1Department of Chemical Engineering, Ariel University, Ariel 4070000, Israel; avitals@ariel.ac.il (A.S.); kossenkoa@ariel.ac.il (A.K.); zinigrad@ariel.ac.il (M.Z.); 2Physics Department, Faculty of Natural Sciences, Ariel University, Ariel 4076414, Israel; viktorde@ariel.ac.il

**Keywords:** plasma electrolytic oxidation, titanium alloys, cleaning, hydroxyapatite, molten salt, organic contamination

## Abstract

The effect of various cleaning methods on coating morphology and their effectiveness in removing organic contaminants has been studied in this research. Bioactive coatings containing titanium oxides and hydroxyapatite (HAP) were obtained through plasma electrolytic oxidation in aqueous electrolytes and molten salts. The cleaning procedure for the coated surface was performed using autoclave (A), ultraviolet light (UV), radio frequency (RF), air plasma (P), and UV-ozone cleaner (O). The samples were characterized using scanning electron microscopy (SEM) with an EDS detector, X-ray photoelectron spectroscopy (XPS), X-ray phase analysis (XRD), and contact angle (CA) measurements. The conducted studies revealed that the samples obtained from molten salt exhibited a finer crystalline structure morphology (275 nm) compared to those obtained from aqueous electrolytes (350 nm). After applying surface cleaning methods, the carbon content decreased from 5.21 at.% to 0.11 at.% (XPS), which directly corresponds to a reduction in organic contaminations and a decrease in the contact angle as follows: A > UV > P > O. This holds true for both coatings obtained in molten salt (25.3° > 19.5° > 10.5° > 7.5°) and coatings obtained in aqueous electrolytes (35.2° > 28.3° > 26.1° > 16.6°). The most effective and moderate cleaning method is ozone treatment.

## 1. Introduction

Titanium and its alloys are the most widely used material in biomedical applications due to their excellent properties, such as high corrosion resistance, high specific strength, and low density [1,2,3]. Artificial joints, dental implants, and orthopedic/bone screws are common application fields for titanium and its alloys [4,5]. Since Ti is bioinert [6], it does not form strong bonds with soft and hard tissues, negatively affecting its bioactive and osseointegration properties and sometimes complicating its use in biomedical applications. To improve these properties various surface modification methods are employed to enhance these properties [7,8], aimed at increasing the contact surface of biological tissue with the implanted material (sandblasting, chemical etching, etc.). As combined methods for improving the surface properties of implanted materials, the sol-gel method, plasma spraying, anodizing, PVD, and CVD are used [9,10]. The combination of these methods enhances the specific surface area of the implanted material and introduces bioactive properties by incorporating various precursors into the coating composition. The introduced precursors usually consist of various combinations of calcium and phosphorus compounds (hydroxyapatite, tricalcium phosphate, etc.) [11,12,13]. Therefore, the modified surface is characterized by higher bioactivity and, as a result, an increased rate of osseointegration [14,15,16,17]. Bioactive coatings based on calcium and phosphorus compositions are utilized because they form compounds that closely resemble the chemical composition of human bone tissue [15,18]. However, it is crucial to remember that the applied coating must meet several strict requirements to ensure its safe usage in the human body. These requirements include excellent corrosion resistance and strong adhesive properties between the resulting coating and the base material of the implant [19,20,21].

The above surface modification methods do not quite meet the stated requirements in the first place because they have relatively low adhesion to the base material. Among other things, due to the use of high-temperature surface modification methods (plasma spray, sol-gel), phase transformations of the base material can occur, altering its mechanical properties [22,23,24]. The anodizing method does not provide good corrosion properties to the surface [25,26]. As a result, the coating may dissolve under the influence of biological fluids, and the implanted material may fall out. Physical and chemical vapor deposition methods (PVD and CVD) have a comparatively low coating rate [27,28] and a complex technological design.

The most promising coating method devoid of the listed disadvantages is Plasma electrolytic oxidation (PEO) in aqueous electrolytes [29,30,31,32] and nitrate salt [33]. The use of molten nitrate salt as a PEO electrolyte is our development, which eliminates the need for electrolyte cooling and makes it possible to reduce unwanted organic and in-organic contaminations in the coating composition. This universal method makes it possible to obtain given chemical and phase composition coatings, for example, for bioactive and anti-corrosion applications on titanium [34] and aluminum [35] alloys. The technological foundations of this method originate from the anodizing process. However, the resulting coatings have a diverse morphology and phase composition. In the PEO process, at the initial stage of coating formation, the process is similar to anodizing. Then, with increasing voltage, micro-arc breakdowns occur, transforming the amorphous structure into a crystalline structure with a ceramic-like coating [36,37]. One of the essential advantages of this process is that the coating is not applied to the surface but grows from within [38], i.e., the coating’s source of the “building material” is the metal itself. Bioactive additives in the form of hydroxyapatite are introduced into the electrolyte and are incorporated into the coating during the PEO process. Depending on the composition and concentration of precursors introduced into the electrolyte, the physical and chemical properties of the coating can be varied and controlled.

Despite the many advantages of the PEO coating, the problem of organic contaminants frequently found on bioactive surfaces still needs to be addressed. The purpose of the cleaning procedure is to remove or reduce the number of contaminants, including hydrocarbons, carboxylates, and debris, on the surface of the substrate [39]. Given the importance of the problem and the fact that we have not found a detailed solution to it in the literature, it seems necessary and useful to compare the level of cleaning the coating from organic contaminants using different surface cleaning methods, each of which has its own significance and different mechanisms of action. Depending on the material’s properties, carefully selecting the methods used for a particular device is necessary. In this study, autoclaving (A) [40], cold plasma (P) [41], ultraviolet radiation (UV) [42], and ozonation (O) [43] were used to clean the coatings obtained on the Ti-6Al-4V alloy from organic contamination. Autoclaving (A) is a physical cleaning method in which organic contaminants are subjected to a certain temperature, pressure, and time (200 °C, 120 min). However, depending on the density, volume, and size of the material, the efficiency of the autoclaving process may vary [44].

In the Air RF plasma, ions are formed, which sputter inorganic material from the surface, and free radicals are formed, which react with and remove organic contaminants. Active species such as polar groups break down and peel off the surface layer. The thickness of the removed layer and the surface properties can be changed depending on the electromagnetic field strength, type of gas, gas pressure and flow rate, plasma mode, and system geometry.

UV light irradiation with a 100–280 nm wavelength has been used to break down organic molecules and clean the surface. The basic principle of the UV cleaning method is to reduce organic contaminants [45] into harmless H_2_O, CO_2_, and NO_x_, which are then effectively removed by rinsing with water.

The UV-ozone method is effective for dry, non-acid, non-destructive atomic cleaning and removal of organic contaminants. A U lamp irradiates two types of high-intensity wavelengths (185 and 254 nm. In the presence of oxygen (airflow), 185-nm irradiation dissociates molecular oxygen O_2_ into triplet atomic oxygen O(^3^P), which combines with molecular oxygen O_2_, generating ozone O_3_. On the other hand, 254-nm UV light dissociates ozone O_3_ and forms molecular oxygen O_2_ and singlet atomic oxygen O(^1^D), which has strong oxidation power and reacts with the sample surfaces. In the case of organic materials, chain scission of molecules happens, and organic residue contaminants are gently removed from the substrates as volatile byproduct molecules such as CO_2_, H_2_O, and O_2_ [46].

Figure 1 schematically shows the process of cleaning the biomaterial’s surface and the techniques used within the framework of the presented work. 

This work aimed to study the effectiveness of various surface cleaning methods of a bioactive coating from organic contaminants and their effect on morphology, phase, and chemical composition, as well as its wettability. The morphology, phase, and chemical composition of the coatings were examined using scanning electron microscopy (SEM), X-ray diffraction (XRD), and X-ray photoelectron spectroscopy (XPS), respectively. The wettability properties of the surface were investigated using Hank’s solution by the sessile drop method with contact angle measurements.

## 2. Materials and Methods

### 2.1. Experimental

Samples of titanium alloy Ti-6Al-4V with the chemical composition are given in Table 1 and were sanded with sandpaper (grit 600–1200), followed by ultrasonic cleaning in acetone. The geometric surface area of the samples was 0.16–0.17 dm^2^.

The electrical parameters of the MP2-AS 35 power supply (Magpulls, Sinzheim, Germany) were pulsed (unipolar square wave—UP+) with a duty cycle of 50% at 300 Hz. They recorded using a Fluke Scope-Meter 199C (200 MHz/2.5 GS s^−1^). The device was located in the electrical circuit between the working container and the power source.

A molten salt PEO process (Figure 2) approach was carried out at 280 °C in an electrolyte composition of 45.7 wt.% NaNO_3_ (Sigma-Aldrich Co., St. Louis, MO, USA) and 54.3 wt.% KNO_3_ (Spectrum Chemical Mfg. Corp., New Brunswick, NJ, USA). The electrolyte was in a nickel crucible (99.95%, Scope Metals Group Ltd., Holon, Israel), which also served as a counter electrode (CE). The anode/cathode surface ratio was 1/30, the anode current density was 4 A/dm^2^, and the voltage was process limited. A fine powder of Ca_10_(PO_4_)_6_(OH)_2_ (Sigma–Aldrich Co., USA) was used as a source of Ca and P with a –concentration maintained in the electrolyte (C = 10 g/kg).

The PEO process in an aqueous solution (Figure 2) was carried out in an electrolyte containing calcium glycerophosphate (C_3_H_7_CaO_6_P, Spectrum Chemical Mfg. Corp.) 12 g/L and calcium acetate (Ca(CH_3_COO)_2_ (Spectrum Chemical Mfg. Corp.) 40 g/L at 30 °C in 400 mL water-cooled glass jar. The Ti-6Al-4V samples were used as a working electrode (WE), and a stainless steel sheet was used as a counter electrode (CE). The anode current density was set at 10 A/dm^2^, and the voltage was limited by the nature of the process. Figure 2 provides a schematic representation of the PEO setup in aqueous electrolytes and molten salts.

Finally, the samples were washed with distilled water and dried using compressed air. Subsequently, the samples underwent hydrothermal treatment (A) for two hours at 200 °C. The treatment was carried out in a stainless steel high-pressure reactor (Berghof BR–100, Berghof GmbH, Eningen Achalm, Germany) using an alkaline solution (pH = 11) of KOH (Sigma–Aldrich Co., St. Louis, MO, USA). This step served to convert the phosphorus and calcium dopants into the hydroxyapatite form and also to remove any organic contaminants from the surface. After the autoclave treatment, three additional cleaning methods were applied to the samples.

One of the methods involved treating the samples with Air RF plasma (P) treatment using the EQ-PCE-3 plasma cleaner (MTI, Yeosu, South Korea) under the following conditions:-The coil’s applied current frequency was 13.56 MHz.-The RF power used was 18 W, and the plasma chamber’s working gas pressure (Air) was 33 Pa.

The SC-UV-I ozone cleaner (Setcas electronics, San Diego, CA, USA) was used for UV and UV-ozone cleaning. The grid lamp emitted wavelengths of 185 and 254 nm, with an intensity of 30 mW/cm^2^ at a 5 mm distance. The samples were positioned 5 cm away from the lamp. Dry air was supplied from a cylinder at an 80 mL/min flow rate for the UV-ozone treatment. The UV treatment was conducted under the same conditions but without the air supply. The treatment time for all the methods above was 5 min.

The codes of experimental samples after PEO and their cleaning are given in Table 2. 

### 2.2. Surface Characterization

The treated samples’ chemical analysis and elemental mapping were conducted using the SEM technique (TESCAN MAIA3, Brno, Czech Republic) equipped with EDS X-MaxN (Oxford Instruments, Abingdon, UK). Phase analysis was performed using XRD (SmartLab SE, Rigaku, Akishima, Japan) with Cu-Kα radiation (λ = 1.5460 Å). Measurements were made in 2Θ geometry (angle of incidence 3°) in the range of 20–80° with a step of 0.03° and a rate of 0.5°/min. The phase analysis was performed using SmartLab Studio II ver. 4.2.44.0 with Powder XRD module and ICDD PDF-2 2019 database. Quantification was performed using the Rietveld Refinement method in the range of 20–80° using the WPPF (Whole Powder Pattern Fitting) module. The crystallite size was determined using the Comprehensive analysis module using the Halder-Wagner modeling method.

X-ray photoelectron spectroscopy (XPS) analysis was carried out using a Nexsa G2 spectrometer (Thermo Fisher Scientific, Oxford, UK) equipped with a monochromated Al Kα X-ray radiation source (photon energy 1486.6 eV, 72 W).

The contact angle (CA) measurements were performed by the sessile drop method using a Hanks’ solution at a temperature of 37 °C and a drop volume of 6 µL. The CAs were measured using a homemade goniometer consisting of an instrument table and a high-resolution digital camera with a maximum magnification of ×1600. The average CA value and standard deviation were calculated from the measurements of five drops on each sample.

After the cleaning process, the surface contaminations of the samples were analyzed using ImageJ ver. 1.53 m software (National Institutes of Health, Bethesda, MD, USA). The analysis was based on the surface morphologies observed in SEM images. For each sample under study, three regions with an area of 0.5 cm^2^ were arbitrarily chosen. 

## 3. Results and Discussion

### 3.1. Plasma Electrolytic Oxidation Processing

The current and voltage plots of the PEO process in aqueous solution (AS) and molten salt (MS) on samples of Ti-6Al-4V alloy are shown in Figure 3.

The voltage-time graphs illustrate the two general steps of the coating formation process. In the first step, a layer of adsorbed ions from the electrolyte composition forms on the surface of the sample [47]. Subsequently, the anodizing process begins, leading to the formation of an amorphous oxide layer [48]. As the oxide layer thickness increases, the required voltage for ion migration through the growing coating also increases. Concurrently, there is a sharp voltage increase accompanied by a drop in current, indicating a change in the dielectric characteristics of the formed coating and a decrease in its conductivity [47].

The formation of an amorphous oxide coating in an aqueous electrolyte can occur up to a voltage of 106 V, whereas in molten salts, it reaches up to 30 V. Figure 3 demonstrates that the currents and voltages required for formation differ by nearly threefold between an aqueous electrolyte and molten salts. This difference is influenced by factors such as the composition of the electrolyte, its temperature, and its conductivity.

The next stage of coating formation is characterized by the formation of avalanche-like sparks on the surface of the processed samples [49], known as micro-arc breakdowns. The appearance of micro-arc breakdowns is attributed to a sharp inflection in Figure 3, and the inflection point (Vs) corresponds to the breakdown of the previously formed amorphous dielectric layer. With the appearance of micro-breakdowns, sparks are formed at specific points, reaching temperatures up to 10,000 K. Consequently, the previously formed amorphous coating undergoes phase transformations, resulting in the formation of crystalline titanium oxide [50]. Additionally, at the moment of breakdown, electrolyte components are drawn into the discharge channel, leading to the formation of additional coating phases and the doping of the coating with the desired components [51]. In our case, these components are calcium and phosphorus, which can form the hydroxyapatite (HAP) phase, thereby promoting the osseointegration and fusion of the implant with the bone matrix [52].

### 3.2. Microstructural Characterization and Chemical Analysis 

SEM images of the samples obtained by the PEO method in aqueous solutions (Figure 4a–d) and in molten salts (Figure 4e–h) and processed by the A method, which, on the one hand, forms crystalline hydroxyapatite from the previously introduced Ca and P precursors in the coating composition [53]. On the other hand, the surface is cleared of organic pollutants by the A method. However, as seen from Figure 4a,e, and Table 3, the A method is insufficient for cleaning the surface of AS-A and MS-A samples, and additional post-processing is required. According to Figure 4, the area of surface contamination after cleaning by the A method for samples AC-A is 955 µm^2^, and for MC-A = 484 µm^2^, and the total contamination over the entire area (0.17 dm^2^) of the samples is, on average, 7.9% and 4%, respectively.

The UV treatment has been applied as an alternative method for more efficient cleaning of organic contaminants. After UV treatment (Figure 4b,f), the amount and size of organic contaminants decreased to 105 μm^2^ for the AS-A-UV sample, which is 0.83% of the total sample area (0.17 dm^2^). For the MS-A-UV sample, the area of residual contamination was 302 μm^2^, corresponding to 2.48% of the total area of the treated surface (0.17 dm^2^). It should also be noted that UV irradiation cleans the surface more effectively than the A method. However, it also contributes to the degradation and destruction of the target coating, associated with a decrease in the size of the HAP crystals (Figure 5b,f and Table 3 and Table 4). This surface reaction can be attributed to the photocatalytic properties of titanium oxide [54,55], which, when interacting with hard ultraviolet light, absorbs it and converts it to thermal energy or singlet oxygen and other radicals. The resulting compounds can enter into successive reactions that lead to the degradation and destruction of the film. The chemical analysis of the surface carried out below using EDS (Table 3) and XPS (Table 4) methods shows a decrease in the surface concentration of calcium and phosphorus, which also indicates the destruction of the coating.

The cold plasma treatment of the coating (Figure 4c,g) did not significantly affect the morphology and chemical composition (Table 3 and Table 4), with a significant reduction in surface contamination can be seen. For example, for sample AS-A-P, the contaminated surface area decreased to 120 μm^2^ (≈1% of the total sample area), and for sample MS-A-P = 205 μm^2^, which corresponds to ≈1.65% of the total sample area 0.17 dm^2^. However, for complete cleaning of the coating, either additional treatment or an increase in the processing time by the P method is required.

The ozone cleaning method was used as an alternative to the methods above. The results presented in Figure 4d,h indicate that ozone treatment does not damage the coating and is the most effective method for dealing with organic contaminations. Surface morphology analysis (Figure 4d) and chemical analysis (Table 3 and Table 4) demonstrate that the AS-A-O sample was completely free of organic contaminants. On the MS-A-O sample, the residual contamination area was 8.1 μm^2^, which accounted for 0.66% of the total sample area (0.17 dm^2^). The cleaning methods were replicated three times, yielding similar results. The organic contaminants exhibited similar size and shape across all the studied samples.

For a more detailed analysis of the coating morphology after cleaning, higher magnification SEM photographs were taken, as well as elemental mapping of the studied coatings (Figure 5). 

During the hydrothermal treatment after PEO in aqueous solutions and molten salt on the treated surface grew HAP crystals with sizes from 0.5 to 3 µm and 0.1 to 1 µm, respectively. The crystals have a regular shape without surface defects (Figure 5a,e). In the case of intense ultraviolet treatment, as mentioned earlier, the surface was severely damaged on a macroscale (Figure 4b,f). At higher magnification, it can be seen that the surface of the coating has degraded, and instead of acicular HAP, the surface has a looser structure (Figure 5b,f). Plasma treatment had a similar effect on the surface of the coating; namely, the HAP crystals became thinner, and their number decreased (Figure 5c,g). It should also be noted that with all the advantages of this method, it is not selective and knocks out both organic contaminants from the surface of the processed material and other target components necessary to impart special properties to the coating. The most effective method of cleaning from organic contaminants in the macro (Figure 4d,h) and micro (Figure 5d,h) scales of all presented is ozone treatment since it removes a significant amount of organic contaminants without destroying the surface layers of the coating and without changing its microstructures [56].

An analysis of the cross-sectional morphology of samples obtained by PEO and treated with A, UV, P, and O is presented in Figure 6.

The morphology of the cross sections of samples obtained in an aqueous electrolyte (Figure 6a–d) has a thickness of 8–10 μm and is a porous structure with a pore size of 0.1–4 μm. The samples obtained in molten salts (Figure 6e–h) have a denser and non-porous coating structure with an average thickness of 1.5–2 μm. As can be seen from the presented micrographs, the surface treatment (A, UV, P, and O) of the coatings to clean them from organic contaminants did not affect their structural integrity. The presented morphological features of the cross-sections of the studied samples are typical. They are determined by the temperature, electrolyte type, voltages, and currents of coating formation, as well as the “nature” of the process itself. 

The chemical composition of the surface Figure 3 measured by the EDS detector is presented in Table 3.

XPS chemical analysis was performed to improve the detection ability and accuracy of the research. The low-resolution spectrum for all samples under study is shown in Figure 7.

XPS analysis confirmed the presence of carbon, oxygen, titanium, calcium, and phosphorus bonds on the surface. The samples obtained by the PEO method in aqueous electrolytes present high contamination with organic substances since organic salts (calcium glyceryl phosphate and calcium acetate) were used as electrolyte components. The samples obtained in molten salts had less organic contamination since molten salts at a temperature of 280 °C were used as an electrolyte.

After surface treatment by hydrothermal treatment, the carbon concentration on the surface was 5.21 ± 0.08 at.% and 4.63 ± 0.06 at.% for samples AS-A (Figure 7a) and MS-A (Figure 7e), respectively. The UV-treated samples showed a higher surface finish, and the carbon concentration was 2.13 ± 0.02 at.% and 1.63 ± 0.07 for AS-A-UV (Figure 7a) and MS-A-UV (Figure 7f), respectively. A fairly good result was shown by the CP method (AS-A-P = 1.01 ± 0.1 at.% and MS-A-P = 0.86 ± 0.2 at.%). However, an organic pollutant remained on the surface, which in the future could interfere with the effective osseointegration of the bioactive surface [57]. It can also cause unwanted inflammatory processes in the human body.

Ozone treatment was the most moderate and effective surface cleaning method since the formed coating, and HAP crystals almost did not change their morphology. The amount of organic contaminants decreased by almost 50 times compared with the samples after A and amounted to 0.11 ± 0.03.% (AS-A-O) and 0.12 ± 0.02 at.% (MS-A-O). The percentage of chemical elements included in the coating increased (Table 4) in proportion to the amount of carbon loss after cleaning.

Based on the results of chemical analysis conducted using EDS (Table 3) and XPS (Figure 7, Table 4), it was observed that during the surface cleaning process, not only were the organic contaminants removed, but the bioactive coating layer was also affected and removed. For example, after UV treatment, the surface concentration of calcium and phosphorus in the coating on samples AS-A-UV and MS-A-UV decreased by almost 30% compared to AS-A and MS-A, respectively. The surfaces of the samples treated with plasma (AS-A-P, MS-A-P) were less destructive compared to the UV method, but the decrease in the concentration of Ca and P was about 17% compared to the samples treated with the A method. Ozone was the most gentle treatment method, which, on average, reduced the surface concentration of bioactive elements (Ca, P) by 3% compared to the base samples AS-A, and MS-A. The results of chemical analysis (EDS, XPS) show a decrease in the concentration of organic impurities or their complete removal, as well as a reduction of the concentration of the near-surface layer of calcium and phosphorus, also confirmed by the morphology of the coating.

It is important to note that all cleaned surfaces exhibit bioactive properties as they contain HAP crystals on their surfaces. Their bioactivity and osseointegration rate is determined by the presence of HAP on the surface and by their size and quantity per surface unit. Thus, with a decrease in the size of HAP crystals and an increase in their number per unit surface, their mechanical strength and osseointegration rate increase [58], demonstrated by coatings obtained in molten salts (Figure 5e–h). 

### 3.3. Phase Characterization

The phase compositions of the oxide layers after cleaning were analyzed using the X-ray patterns presented in Figure 8.

The phase composition of the coatings obtained by the PEO method in aqueous electrolytes and molten salts after surface cleaning of organic contaminants was determined by XRD analysis. The coatings consist of anatase, rutile, and HAP phases. As a result of phase analysis, it was revealed that, regardless of the method of coating preparation (aqueous electrolytes or molten salts), surface cleaning methods have little effect on the final phase composition of the coating. However, it should be noted that based on the quantitative phase analysis using the Rietveld Refinement method (Table 5), the content of the HAP phase in the samples obtained in molten salts is almost 2.5 times higher than in the samples obtained in aqueous electrolytes. These results are further supported by the surface morphology observations (Figure 5), which reveal that the HAP crystals obtained in molten salts are smaller (Figure 5e) and more evenly distributed across the surface. Additionally, their “packing” number and density are significantly higher than the HAP obtained in aqueous solutions (Figure 5a). Furthermore, the content of the anatase phase in samples obtained in aqueous electrolytes is almost three times higher than in molten salt samples.

According to XRD data for coatings obtained in aqueous electrolytes and molten salt, shown in Figure 8, the crystallite size (D) was calculated using the Debye–Scherrer equation:
(1)
$$D=\frac{K\cdot \lambda }{\beta \cdot cos\theta }$$

where: β refers to the half-width of the detected diffraction maximum. The parameter K, commonly referred to as the shape factor, typically has a value of approximately 0.9, and λ denotes the wavelength of the X-ray source.

Based on the calculated data, it was revealed that the coating obtained in aqueous electrolytes consists of anatase (D = 85–95 Å), rutile (D = 86–177 Å), and HAP (D = 270–354 Å) crystallites. 

The crystallite size obtained in molten salts by the PEO method for anatase (143–172 Å) and rutile (243–278 Å) increased by 1.5–2 times compared to the results in aqueous electrolytes. Most likely, the coatings obtained in molten salts have a larger crystallite size since the electrolyte used is kept at a higher temperature (280 °C) [59]. The increased temperature promotes the growth of crystallites. In addition, it is pertinent to highlight that the elevated temperature of the electrolyte has a positive effect on the formation of coatings with high crystallinity, which is often difficult to achieve in aqueous electrolytes. Typically, the phase composition of coatings obtained in an aqueous electrolyte consists of a mixture of amorphous and crystalline phases. The size of the formed HAP crystallites included in the composition of the coating obtained in an aqueous electrolyte and molten salts have similar dimensions. It should also be noted that during the cleaning process, the size of HAP crystallites decreases (Figure 9, Table 5) in the following sequence for coatings obtained in aqueous electrolytes AS-A (354 Å) > AS-A-O (323 Å) > AS-A-P (275 Å) > AS-A-UV (270 Å) and molten salts MS-A (278 Å) > MS-A-O (270 Å) > MS-A-P (260 Å) > MS-A-UV (243 Å). This pattern of degradation of the HAP coating is also confirmed by the results of the surface morphology analysis (Figure 4 and Figure 5) and its chemical composition (Figure 7 and Table 3 and Table 4). An opposite relationship is observed for the anatase and rutile phases, i.e., during the cleaning process, crystallites grow in the following sequence for anatase AS-A (85 Å) < AS-A-O (88 Å) < AS-A-P (90 Å) < AS-A-UV (94 Å), MS-A (143 Å) < MS-A-O (155 Å) < MS-A-P (160 Å) < MS-A-UV (172 Å), as well as for rutile AS-A (86 Å) < AS-A-O (87 Å) < AS-A-P (149 Å) < AS-A-UV (177 Å), MS-A (243 Å) < MS-A-O (263 Å) < MS-A-P (273 Å) < MS-A-UV (278 Å). As a result, it can be concluded that the ozone surface treatment method most effectively fights organic contaminations (Figure 4d,h and Figure 7d,h) and practically does not affect the crystallinity (Figure 9), phase (Figure 8), chemical (Figure 7) and morphology (Figure 5d,h) composition of the coatings.

### 3.4. Wettability

One of the main conditions for successful dental implantation is sufficient primary stabilization of the implant in the bone tissue [60]. Primary stability depends on the geometry of the implant and its topography [61]. The presence of micro defects (pores) [62] and bioactive material (HAP) crystals on the implant surface increases the contact area. It allows osteoblasts to attach the implant surface using structural proteins and glycoproteins more effectively, thereby contributing to the acceleration of the osseointegration process [63]. The effects of Facilitating the process of osseointegration are related to the surface’s wettability. 

The study of surface wettability was carried out for samples obtained after PEO treatment in an aqueous solution and molten salts, followed by their purification by various methods presented above. Wettability was examined by measuring the contact angle (CA) using Hank’s solution kept at 37 °C.

Dripped drops of Hanks’ solution and the measured CA’s are shown in Figure 10 and Table 6.

The untreated Ti-6Al-4V alloy had a less hydrophilic surface with CA = 78.2 ± 1.1°, compared with the treated samples in Table 6.

After the PEO, in the process of surface treatment by the A method, HAP crystals are formed from the calcium and phosphorus precursors previously introduced into the coating composition, increasing the specific surface of the samples. In addition, the A method cleans the surface of organic contaminants. Based on the results of the contact angles measurements (Table 6), the fine-grained structure of HAP obtained in molten salt has more hydrophilic surface properties (MS-A = 25.3 ± 1.1°) compared to the sample obtained in aqueous electrolyte (AS-A = 35.2 ± 1.5°). The hydrophilicity of the coated surface increased with further surface treatment by the UV, P, and O methods, which directly corresponds to a reduction in organic contaminations and a decrease in the contact angle as follows: A > UV > P > O. This holds true for coatings obtained in molten salt (25.3° > 19.5° > 10.5° > 7.5°) and coatings obtained in aqueous electrolytes (35.2° > 28.3° > 26.1° > 16.6°). Analyzing the morphology, phase, and chemical composition was also confirmed by reducing organic contamination. 

An additional and significant factor affecting the surface’s physical properties is the coating’s degradation during cleaning. This pattern was discussed above for both MS and AS coatings.

Based on the results, ozone cleaning is the most effective and moderate surface treatment method. This treatment not only preserves the HAP crystals (Figure 4d,h) but also efficiently eliminates organic contaminants (Figure 5d,h and Figure 7d,h), significantly reducing contact angles. Specifically, the contact angle for sample AC-A-O is reduced to 16.6 ± 0.7°, while for samples MS-A-O, it is reduced to 7.5 ± 0.4°.

This phenomenon is associated with the generation of singlet oxygen during ozone surface treatment. It efficiently eliminates organic contaminants and enhances the formation of polar groups on the surface, thereby increasing its surface energy and resulting in more excellent hydrophilicity.

## 4. Conclusions

In this study, the PEO method was used to produce bioactive coatings on the surface of the Ti-6Al-4V alloy using aqueous electrolytes and molten salt. The coatings consisted of rutile, anatase, and hydroxyapatite (HAP) crystals. The impact of the preparation methods on the coating’s morphology, phase, and chemical composition was investigated. A comprehensive analysis was conducted to compare the efficacy of various cleaning methods for removing organic contaminants. The study established the effects of the cleaning methods on the structure and physicochemical properties of the treated surfaces and their effectiveness in removing organic contaminants. The main conclusions are as follows:The coating obtained using molten salt exhibited a more uniform distribution of hydroxyapatite crystals across the surface than similar coatings produced using an aqueous electrolyte. Additionally, the concentration of hydroxyapatite crystals, as determined by EDS, and the phase composition, as determined by XRD, were found to be 1.5 and 2.2 times higher, respectively, in the coating obtained in molten salt compared to the aqueous electrolyte coating.During the cleaning process of samples from organic contaminants using autoclaving (A), UV, cold plasma (P), and ozone (O), a comprehensive analysis was performed using scanning electron microscopy (SEM), X-ray diffraction (XRD), energy-dispersive X-ray spectroscopy (EDS), and X-ray photoelectron spectroscopy (XPS). Scanning electron microscopy (SEM), X-ray diffraction (XRF), energy dispersive X-ray spectroscopy (EDS) and X-ray photoelectron spectroscopy (XPS) were used for complex analysis of the samples cleaned from organic contaminants using autoclaving (HTO), UV, cold plasma (P), and ozone (O). The results revealed that A, UV, and P treatments exhibited a moderate cleaning capability but also led to the degradation of the bioactive coating. On the other hand, ozone treatment proved to be highly effective in removing organic contaminants without adversely affecting the structural and morphological properties of the coating.The sessile drop method was used to study the hydrophilic properties of the coatings after their cleaning by the A, UV, P, and O methods. The coating obtained in molten salt and treated for organic contaminants by the ozone cleaning method was found to have the highest hydrophilicity (CA = 7.5 ± 0.7°).It has been established that the coatings formed on Ti-6Al-4V alloys by the PEO method in molten salts have an increased uniformity and content of HAP and can be promising for medical implantology, and ozone treatment is an effective and sparing procedure for removing organic contaminants.

## Figures and Tables

**Figure 1 materials-16-04624-f001:**
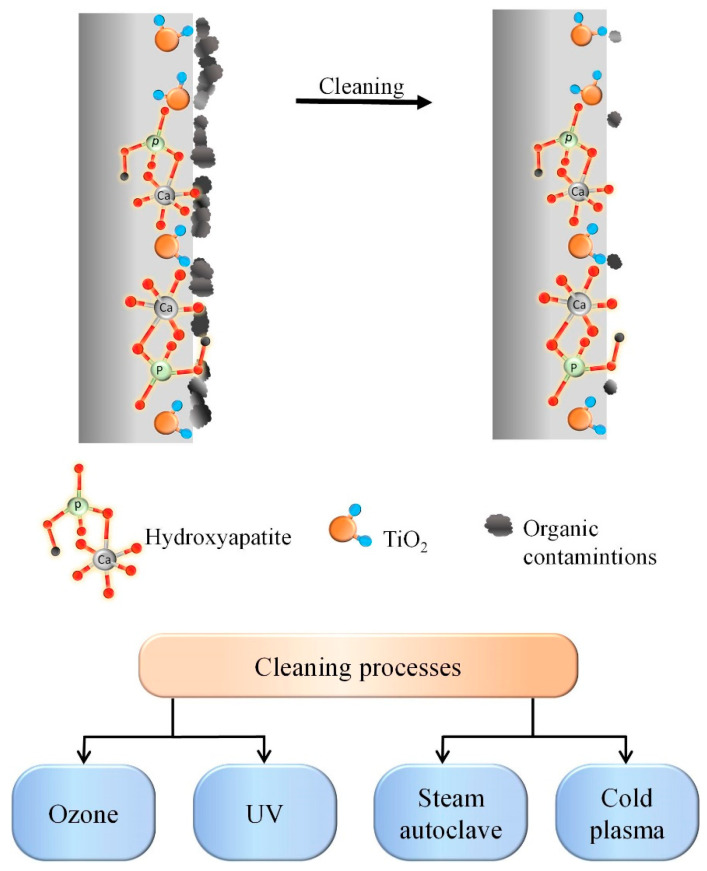
Illustration of surface cleaning of samples after PEO and the methods used in this study.

**Figure 2 materials-16-04624-f002:**
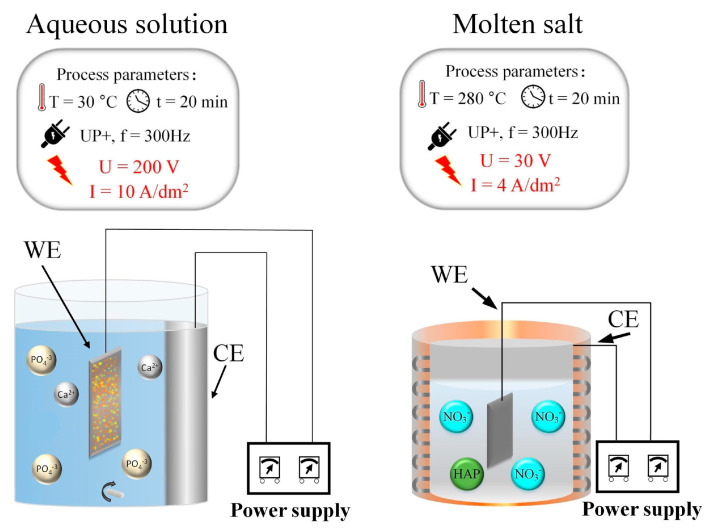
A schematic presentation of the setup for PEO in aqueous electrolytes and molten salts.

**Figure 3 materials-16-04624-f003:**
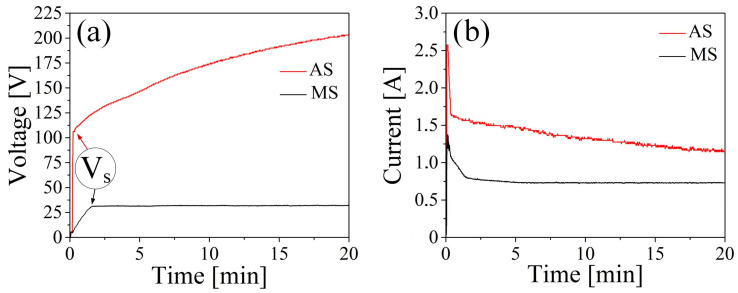
Plots of the PEO treatment in AS and MS electrolyte after 20 min: (**a**) voltage-time curve; (**b**) current-time curve.

**Figure 4 materials-16-04624-f004:**
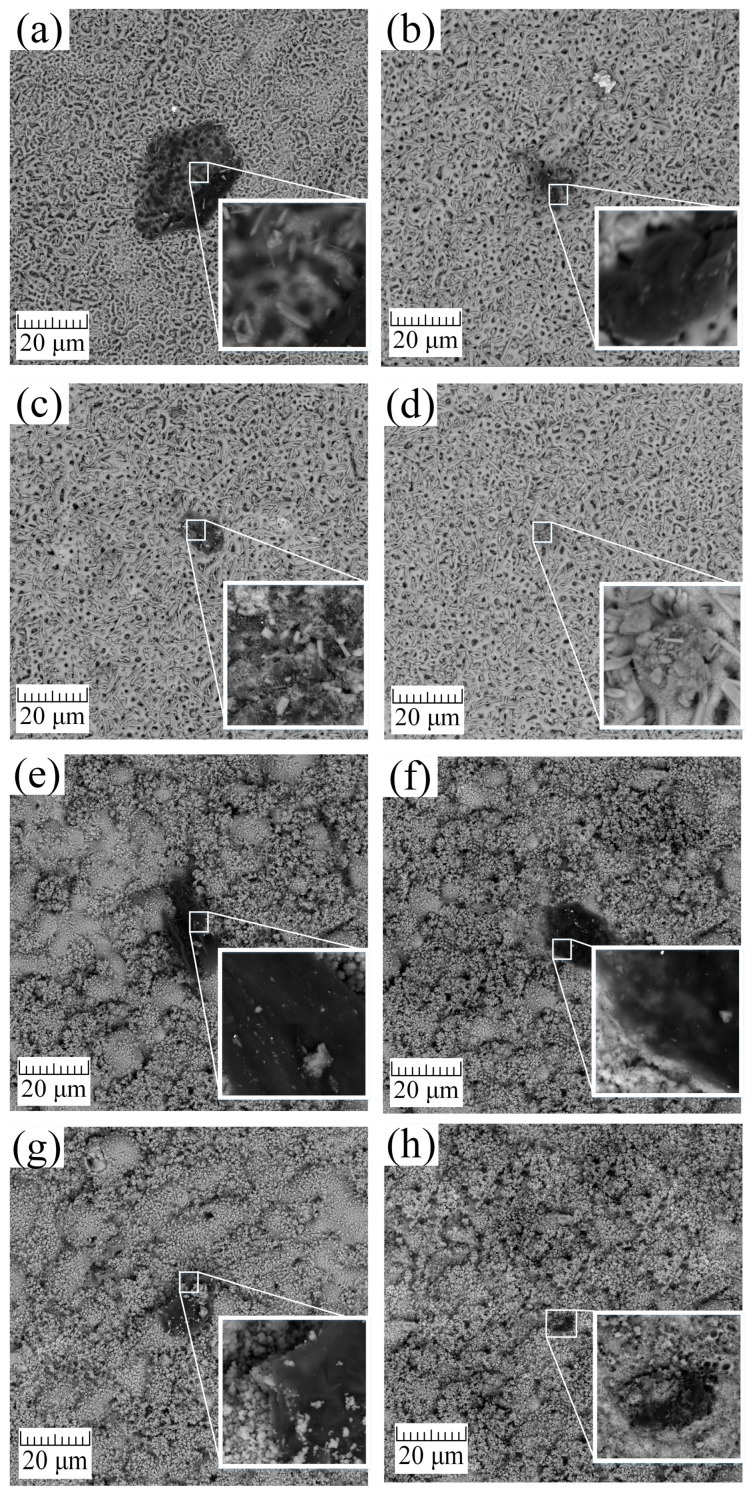
Influence of cleaning procedures on the surface morphology of samples obtained in aqueous electrolytes and molten salts by the PEO method: (**a**) AS-A; (**b**) AS-A-UV; (**c**) AS-A-P; (**d**) AS-A-O; (**e**) MS-A; (**f**) MS-A-UV; (**g**) MS-A-P; (**h**) MS-A-O.

**Figure 5 materials-16-04624-f005:**
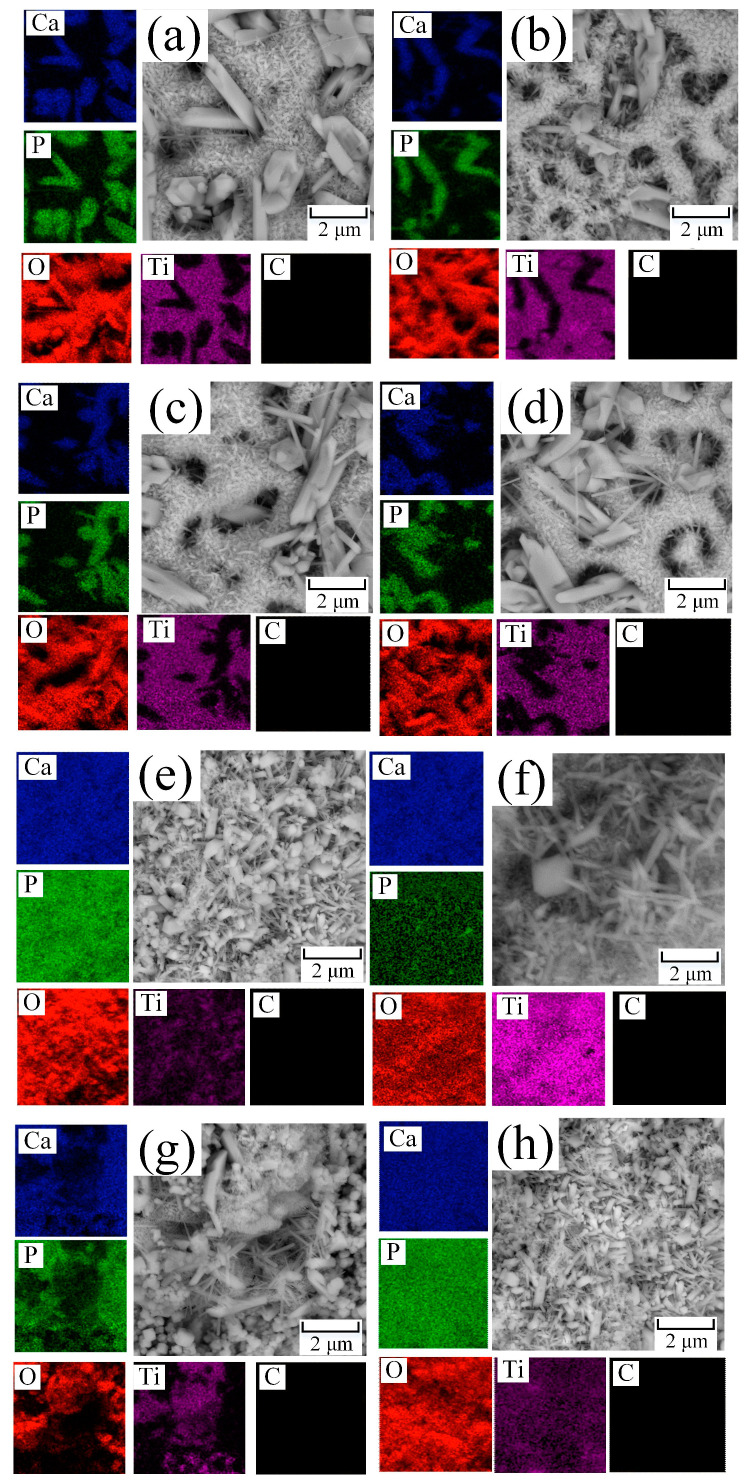
Evolution of surface morphology after cleaning in high resolution: (**a**) AS-A; (**b**) AS-A-UV; (**c**) AS-A-P; (**d**) AS-A-O; (**e**) MS-A; (**f**) MS-A-UV; (**g**) MS-A-P; (**h**) MS-A-O.

**Figure 6 materials-16-04624-f006:**
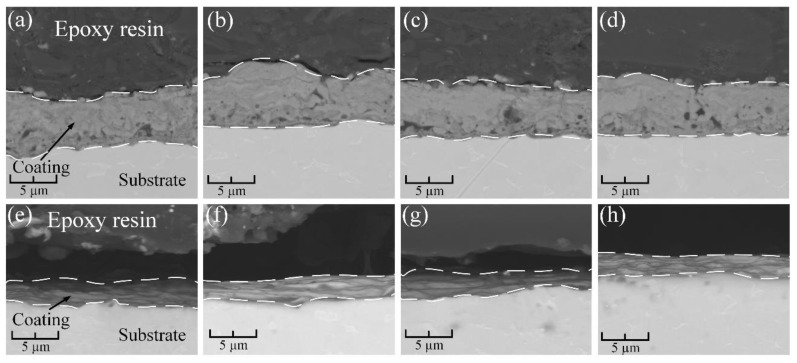
Cross-section morphology for samples after surface cleaning: (**a**) AS-A; (**b**) AS-A-UV; (**c**) AS-A-P; (**d**) AS-A-O; (**e**) MS-A; (**f**) MS-A-UV; (**g**) MS-A-P; (**h**) MS-A-O.

**Figure 7 materials-16-04624-f007:**
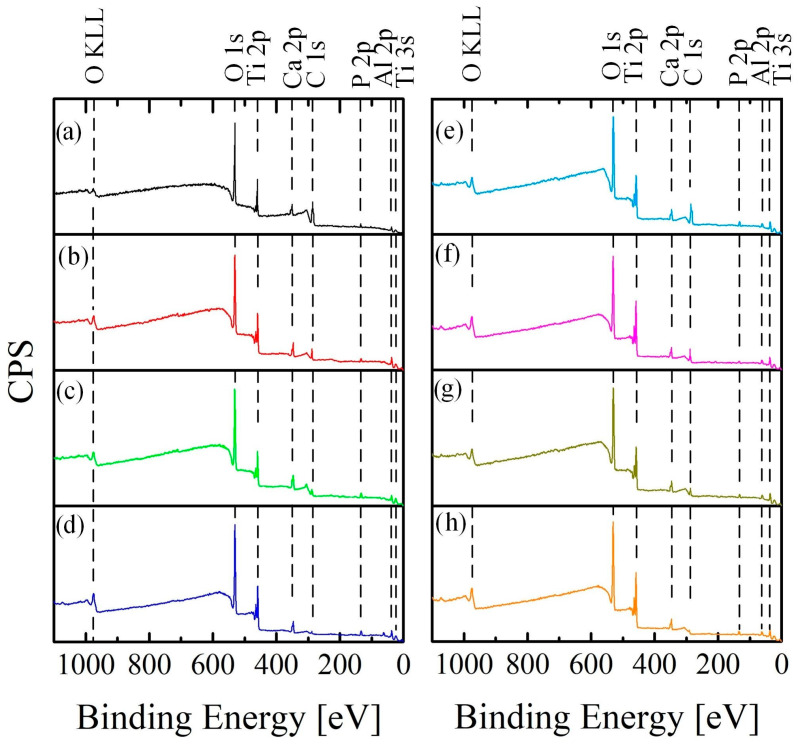
XPS low-resolution spectrum for samples after surface cleaning: (**a**) AS-A; (**b**) AS-A-UV; (**c**) AS-A-P; (**d**) AS-A-O; (**e**) MS-A; (**f**) MS-A-UV; (**g**) MS-A-P; (**h**) MS-A-O.

**Figure 8 materials-16-04624-f008:**
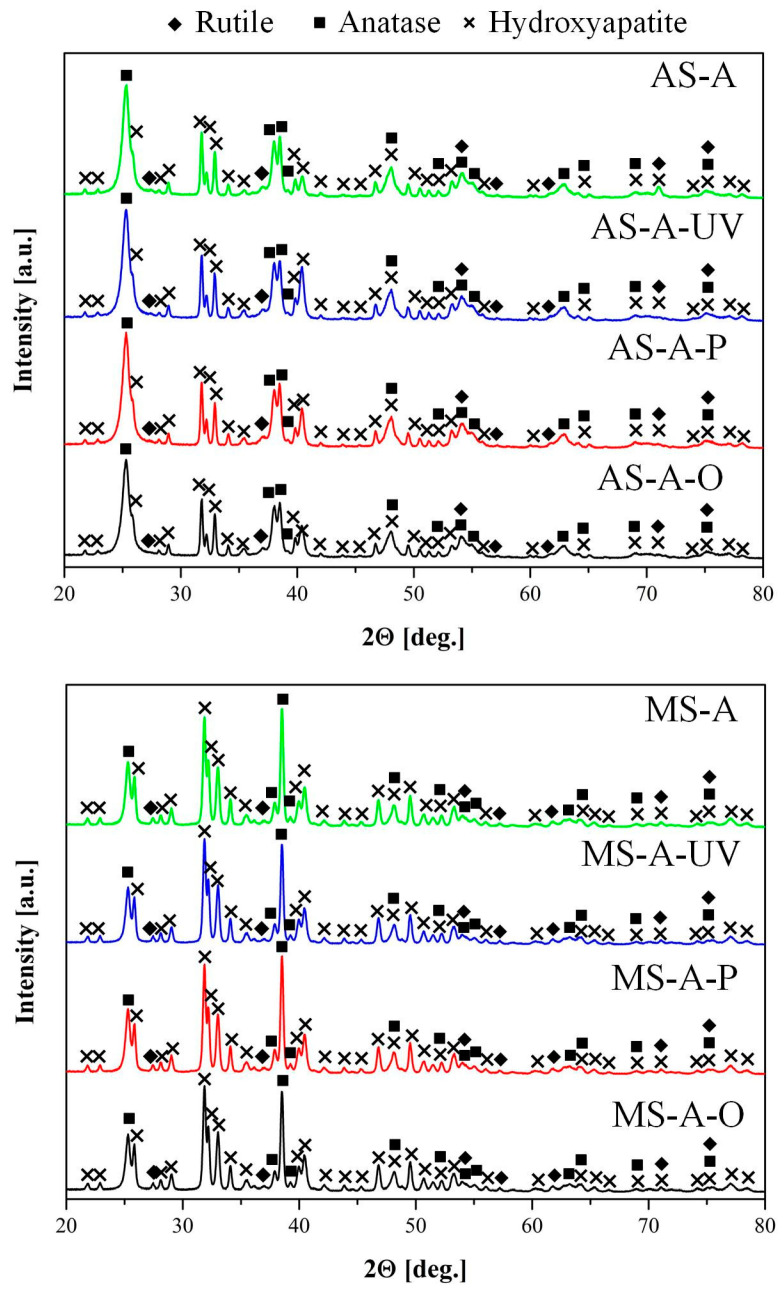
X-ray diffraction patterns after cleaning.

**Figure 9 materials-16-04624-f009:**
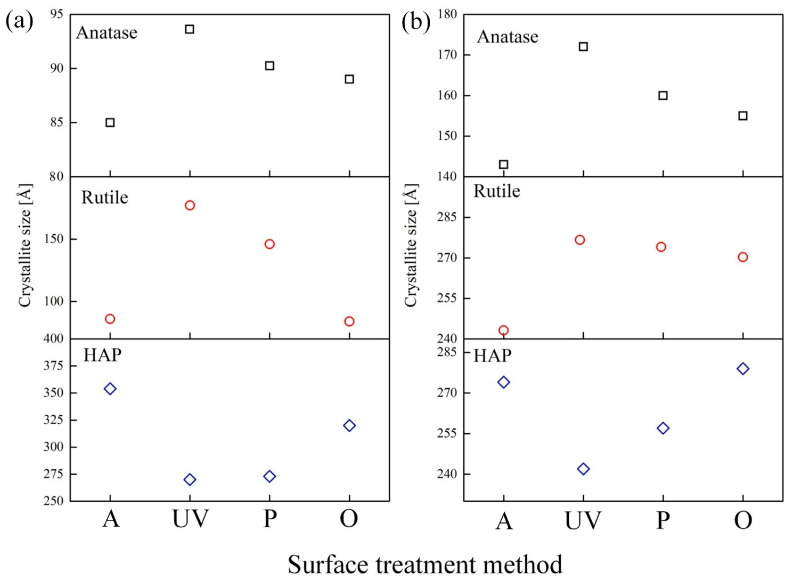
Influence of cleaning methods on the crystallite size of coatings obtained by the PEO method in aqueous electrolytes (**a**) and molten salts (**b**).

**Figure 10 materials-16-04624-f010:**
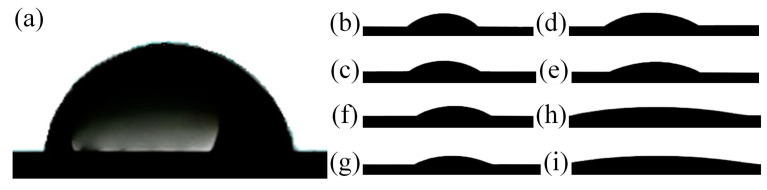
Wettability of (**a**) Ti-6Al-4V; (**b**) AS-A; (**c**) AS-A-UV; (**d**) AS-A-P; (**e**) AS-A-O; (**f**) MS-A; (**g**) MS-A-UV; (**h**) MS-A-P; (**i**) MS-A-O examined with drops of Hanks’ solution at 37 °C.

**Table 1 materials-16-04624-t001:** Chemical composition of the Ti-6Al-4V alloys.

Chemical Element	Fe	V	Al	Ti
Weight %	<0.3	3.5–4.5	5.5–6.5	Balance

**Table 2 materials-16-04624-t002:** Sample codes were examined in this work.

Samples	Treatment Description
AS-A	PEO in aqueous solution + A
MS-A	PEO in molten salt + A
AS-A-UV	PEO in aqueous solution + A + UV 5 min
MS-A-UV	PEO in molten salt + A + UV 5 min
AS-A-P	PEO in aqueous solution + A + cold plasma 5 min
MS-A-P	PEO in molten salt + A + cold plasma 5 min
AS-A-O	PEO in aqueous solution + A + ozone 5 min
MS-A-O	PEO in molten salt + A + ozone 5 min

**Table 3 materials-16-04624-t003:** Elemental composition (at.% by EDS) of the samples in Figure 4 after cleaning.

Sample	Ti	Ca	P	O	C
AS-A	22.9 ± 0.1	6.3 ± 0.1	3.8 ± 0.1	balance	5.3 ± 0.1
AS-A-UV	22.5 ± 0.1	4.3 ± 0.1	2.6 ± 0.1	balance	2.1 ± 0.1
AS-A-P	22.6 ± 0.1	5.7 ± 0.1	3.6 ± 0.1	balance	1.1 ± 0.1
AS-A-O	22.6 ± 0.1	6.2± 0.1	3.6 ± 0.1	balance	-
MS-A	22.4 ± 0.1	9.8 ± 0.1	5.9 ± 0.1	balance	4.7 ± 0.1
MS-A-UV	22.3 ± 0.1	5.9 ± 0.1	3.5 ± 0.1	balance	1.6 ± 0.1
MS-A-P	22.5 ± 0.1	8.2 ± 0.1	4.9 ± 0.1	balance	0.8 ± 0.1
MS-A-O	22.8 ± 0.1	9.6 ± 0.1	5.7 ± 0.1	balance	-

**Table 4 materials-16-04624-t004:** Elemental composition of sample surfaces by the XPS method after cleaning.

Sample	Ti	Ca	P	O	C
AS-A	22.91 ± 0.01	6.32 ± 0.01	3.84 ± 0.01	balance	5.21 ± 0.08
AS-A-UV	22.43 ± 0.01	4.27 ± 0.01	2.63 ± 0.01	balance	2.13 ± 0.02
AS-A-P	22.61 ± 0.01	5.72 ± 0.01	3.61 ± 0.01	balance	1.01 ± 0.01
AS-A-O	22.67 ± 0.01	6.14 ± 0.01	3.65 ± 0.01	balance	0.11 ± 0.03
MS-A	22.44 ± 0.01	9.86 ± 0.01	5.91 ± 0.01	balance	4.63 ± 0.06
MS-A-UV	22.31 ± 0.01	5.93 ± 0.01	3.54 ± 0.01	balance	1.63 ± 0.07
MS-A-P	22.52 ± 0.01	8.22 ± 0.01	4.96 ± 0.01	balance	0.86 ± 0.02
MS-A-O	22.84 ± 0.01	9.62 ± 0.01	5.74 ± 0.01	balance	0.12 ± 0.02

**Table 5 materials-16-04624-t005:** XRD phase composition obtained by Rietveld refinement method.

Sample	Anatase, wt.%	Rutile, wt.%	Hydroxyapatite, wt.%	Rwp, %
AS-A	60.6 ± 0.6	5.9 ± 0.7	33.5 ± 0.5	8.75
AS-A-UV	57.2 ± 1.2	11.7 ± 1.9	31.1 ± 0.8	8.08
AS-A-P	67.4 ± 0.7	0.8 ± 0.7	31.7 ± 0.5	8.31
AS-A-O	54.4 ± 1.9	12.5 ± 1.2	33.1 ± 1.1	10.00
MS-A	21.2 ± 1.2	3.0 ± 0.6	75.8 ± 1.2	11.41
MS-A-UV	22.1 ± 1.6	1.8 ± 0.5	76.1 ± 1.4	9.93
MS-A-P	21.1 ± 1.6	3.4 ± 0.9	75.5 ± 1.6	13.81
MS-A-O	23.1 ± 1.5	2.4 ± 0.7	74.5 ± 1.3	11.45

**Table 6 materials-16-04624-t006:** CA values for Ti-6Al-4V and samples after multi-stage treatment.

Sample	Ti-6Al-4V	AS-A	AS-A-UV	AS-A-P	AS-A-O
CA [°]	78.2 ± 1.1	35.2 ± 1.5	28.3 ± 0.8	26.1 ± 1.4	16.6 ± 0.7
**Sample**		**MS**-**A**	**MS**-**A**-**UV**	**MS**-**A**-**P**	**MS**-**A**-**O**
CA [°]		25.3 ± 1.1	19.5 ± 0.9	10.5 ± 0.4	7.5 ± 0.4

## Data Availability

All the data supporting the findings of this study are available within the article.

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
