# Peer review of "Cleaning Strategies of Synthesized Bioactive Coatings by PEO on Ti-6Al-4V Alloys of Organic Contaminations"

_materials, 2023, doi:10.3390/ma16134624_

Round 1
Reviewer 1 Report
Please see the attached.

Moderate editing of the English language required
Author Response
Reviewer 1
The authors are very grateful to the Reviewer for their valuable comments, which improve the qualitative analysis of the results and give a better understanding of the issue's essence in the present work.
Below we provide the point-by-point responses, and all modifications in the manuscript have been highlighted in yellow:
1) Include some results (numeric) in the abstract section.
Answer:
Thanks for the valuable remark. The following text has been added to the manuscript and highlighted in yellow:
The effect of various cleaning methods on coating morphology and their effectiveness in removing organic contaminants has been studied in this research. Bioactive coatings containing titanium oxides and hydroxyapatite (HAP) were obtained through plasma electrolytic oxidation in aqueous electrolytes and molten salts. The cleaning procedure for the coated surface was performed using autoclave (A), ultraviolet light (UV), radio frequency (RF), air plasma (P), and UV-ozone cleaner (O). The samples were characterized using scanning electron microscopy (SEM) with an EDS detector, X-ray photoelectron spectroscopy (XPS), X-ray phase analysis (XRD), and contact angle (CA) measurements. The conducted studies revealed that the samples obtained from molten salt exhibited a finer crystalline structure morphology (275 nm) compared to those obtained from aqueous electrolytes (350 nm). After applying surface cleaning methods, the carbon content decreased from 5.21 at.% to 0.11 at.% (XPS), which directly corresponds to a reduction in organic contaminations and a decrease in the contact angle as follows: A > UV > P > O. This holds true for both coatings obtained in molten salt (25.3° > 19.5° > 10.5° > 7.5°) and coatings obtained in aqueous electrolytes (35.2° > 28.3° > 26.1° > 16.6°). The most effective and moderate cleaning method is ozone treatment.
2) Replace Fig 4 with high resolution one.
Answer:
Thank you for comment. Figure 4 has been changed.
3) Include a schematic diagram of the composite fabrication process in the experimental section.
Answer:
Thanks for the valuable remark. The following text and Figure 2 has been added to the manuscript and highlighted in yellow:
Figure 2 provides a schematic representation of the PEO setup in aqueous electrolytes and molten salts.
Figure 2. A schematic presentation of the setup for PEO in aqueous electrolytes and molten salts.
4) Line 467, “The untreated Ti – 6Al – 4V alloy had the most hydrophobic surface with CA = 445 78.2±1.1°. The untreated surface is hydrophobic or hydrophilic? The CA is 78.2.
Answer:
Thank you for good comment. The sentence has been rephrased and added to the manuscript.
The untreated Ti - 6Al - 4V alloy had a less hydrophilic surface with CA = 78.2 ± 1.1°, compared with the treated samples in Table 6.
5) There is some clumsy English grammar/phrasing throughout the manuscript. English/ grammar of the manuscript can be improved before publication.
Answer:
Thanks for the valuable remark. A deep correction of the English language was carried out. Some sentences have been rephrased.

Reviewer 2 Report
Please address following concerns/questions:
1. line 78-80: Its not clear on how the purpose of cleaning procedure is to remove blood and proteins. How is the blood come into picture before the implantation into body. Isn't the cleaning is supposed to address before the implant.
2. line 235: Please explain how is the area of contamination calculated on what basis? Is taking 1 SEM image per sample good enough to extrapolate to the whole surface?
3. In the figure4 why is the color code for carbon showing completely dark and indistinguishable from one to another? can you include elemental counts EDS plots.
4. In figure 5, why is the P 2P peak is absent/very low in (a) but showing higher in value table 4. Please explain,
5. what is the theoretical explanation behind better wettability for ozone treatment.
Reviewer 3 Report
The manuscript has the potential for publication, as it provides up-to-date information on the production of biocompatible coatings by plasma electrolytic oxidation. I have a few comments that authors should pay attention:
1. The cross section SEM images of the electrode surface should be provided to determine changes in the electrode surface. Does it happen evenly or not?
2. How does the content of various phases and wettability on the electrode surface change with the time of plasma electrolytic oxidation?
3. Biocompatibility or bioactivity and corrosion in the biosimulated body fluid should also be determined.
4. Figures 3 and 4 are too low quality, the text are not readable, the quality should be increased.
I recommend that this manuscript be published after appropriate revisions have been made.
Round 2
Reviewer 3 Report
In general, I believe that the article has been corrected in accordance with my comments and it can be published. But, I have one remark regarding the design of figure 5. It seems to me that it would be ideal to give this figure as supplimentary materials and present the elemental mapping figures with enlarged size.
Author Response
Dear Reviewer,
Figure 5 with higher resolution and larger pictures added to supporting materials.
Sincerely,
Dr. Alexander Sobolev
